# Neoadjuvant Gastric Cancer Treatment and Associated Nutritional Critical Domains for the Optimization of Care Pathways: A Systematic Review

**DOI:** 10.3390/nu15102241

**Published:** 2023-05-09

**Authors:** Marta Correia, Ines Moreira, Sonia Cabral, Carolina Castro, Andreia Cruz, Bruno Magalhães, Lúcio Lara Santos, Susana Couto Irving

**Affiliations:** 1CBQF—Centro de Biotecnologia e Química Fina—Laboratório Associado, Escola Superior de Biotecnologia, Universidade Católica Portuguesa, Rua Diogo Botelho 1327, 4169-005 Porto, Portugal; 2Portuguese Oncology Institute of Porto (IPO-Porto)—Nutrition, 4200-072 Porto, Portugal; 3Experimental Pathology and Therapeutics Group, Portuguese Oncology Institute of Porto (IPO-Porto), 4200-072 Porto, Portugal; 4Medical Oncology Department, Portuguese Oncology Institute of Porto (IPO-Porto), 4200-072 Porto, Portugal; 5School of Health, University of Trás-os-Montes e Alto Douro (UTAD), 5000-801 Vila Real, Portugal; 6Oncology Nursing Research Unit IPO Porto Research Center (CI-IPOP), Portuguese Oncology Institute of Porto (IPO Porto), 4200-072 Porto, Portugal; 7Surgical Oncology Department, Portuguese Oncology Institute of Porto (IPO-Porto), 4200-072 Porto, Portugal

**Keywords:** nutritional status, nutrition support, nutrition impact symptoms, sarcopenia, neoadjuvant chemotherapy, pre-operative, stomach neoplasms

## Abstract

(1) Background: Gastric cancer patients are known to be at a high risk of malnutrition, sarcopenia, and cachexia, and the latter impairs the patient’s nutritional status during their clinical course and also treatment response. A clearer identification of nutrition-related critical points during neoadjuvant treatment for gastric cancer is relevant to managing patient care and predicting clinical outcomes. The aim of this systematic review was to identify and describe nutrition-related critical domains associated with clinical outcomes. (2) Methods: We performed a systematic review (PROSPERO ID:CRD42021266760); (3) Results: This review included 14 studies compiled into three critical domains: patient-related, clinical-related (disease and treatment), and healthcare-related. Body composition changes during neoadjuvant chemotherapy (NAC) accounted for the early termination of chemotherapy and reduced overall survival. Sarcopenia was confirmed to have an independent prognostic value. The role of nutritional interventions during NAC has not been fully explored. (4) Conclusions: Understanding critical domain exposures affecting nutritional status will enable better clinical approaches to optimize care plans. It may also provide an opportunity for the mitigation of poor nutritional status and sarcopenia and their deleterious clinical consequences.

## 1. Introduction

Gastric cancer is the fifth most commonly diagnosed solid tumor and one of the leading causes of cancer-related deaths worldwide [1].

Gastric cancer patients are known to be at high risk of malnutrition, sarcopenia, and cachexia [2]. Often, malnutrition can be observed at diagnosis [2], and weight loss is also commonly reported at presentation [3]. Evidence has been accumulating to strengthen the adverse influence of an impaired nutritional status on a patient’s clinical course, treatment response [4], and quality of life [5].

Neoadjuvant treatment (NT) encompasses the therapeutic approaches in the immediate period leading to surgery. NT in gastric cancer only includes chemotherapy [6] with the intention to reduce tumor size, increase the possibility of a R0 resection, attempt to treat potential micrometastatic disease, and improve overall survival.

ESMO’s (European Society of Medical Oncology) 2022 guideline, which has been widely adopted in Europe [7], recommends a perioperative chemotherapy regimen with a combination of platinum/fluoropyrimidine for patients with resectable gastric cancer [8]. Following on from the MAGIC [9] and the FFCD/FNCLCC trials [10], the use of ECF (epirubicin, cisplatin, and 5-fluorouracil) or CF (cisplatin and 5-FU), respectively, is common. More recently, the FLOT4-AIO trial showed an increased benefit in the use of the FLOT (fluorouracil, leucovorin, oxaliplatin, and docetaxel) scheme in the perioperative setting [9]. This approach of a fluoropyrimidine-platinum doublet or triplet before surgery is recommended for 2 to 3 months [9]. During neoadjuvancy, most patients are managed at outpatient clinics; hence, it is crucial that this population be best supported to minimize adverse symptoms while remaining in the community. Further, and as a consequence, locally advanced gastric patients have longer care continuums with the prospect of accumulating several nutritional risk exposures along the way, encompassing both disease and iatrogenic impact.

Nutritional status has been shown to strongly impair chemotherapy (CT) success, postoperative prognosis, overall and disease-specific survival (DSS), the rate of complications, and the length of hospital stay.

Thus, a clearer identification and description of nutrition-related critical points throughout neoadjuvant treatment for gastric cancer might be relevant for improving patient care and outcomes.

## 2. Materials and Methods

This systematic review followed the preferred reporting items for systematic reviews and meta-analyses (PRISMA) reporting guidelines. The protocol has also been registered on the International Prospective Register of Systematic Reviews (PROSPERO), the University of York Centre for Reviews and Dissemination PROSPERO, August 2021 (CRD42021266760). Available from: https://www.crd.york.ac.uk/prospero/display_record.php?ID=CRD42021266760, accessed on 8 August 2021.

### 2.1. Sources and Searches

The following databases: Pubmed/Medline, US National Library of Medicine’s PubMed, ISI’s Web of Knowledge, Cochrane, and Scopus databases were systematically searched using the search string (((Gastric OR Stomach) AND (Cancer OR Neoplasm OR Carcinoma OR Malignancy)) AND (Neoadjuvant OR Pre-operatory) AND (Nutritional status OR Nutritional intervention OR Nutritional support OR Dietary counseling OR Oral nutritional supplements)). An example of the search strategy used can be found in Appendix A.

### 2.2. Study Selection

Two reviewers (MC and ICM) screened the studies against the review’s predefined inclusion criteria (Table 1).

The types of studies that were included in this review were randomized clinical trials (RCTs), surveys, and observational studies such as cohort and case-control studies. All disagreements were debated until a consensus was reached with the assistance of a third subsequent reviewer (MC, ICM, and SCI). Fourteen studies were selected for inclusion.

### 2.3. Data Extraction

Data extraction was performed independently by two reviewers (MC and ICM), using a standardized data extraction template, and following the PI/ECO format. The extraction data divergence was resolved by the third independent reviewer (SCI).

## 3. Results

This systematic review included 14 studies (Figure 1), two of which (14.3%) were RCTs and 11 (78.6%) were cohorts, mainly retrospectively assessed; one out of the eleven included cohorts was assessed prospectively (9.1%). More than half (57.1%) of the included studies comprised body composition analysis data using CT scans or ultrasounds (42.9%), followed by nutritional biomarkers or indices (28.6%). Lastly, only three nutrition support studies (21.4%), comprising an immunonutrition and an ERAS protocol, were eligible.

The selected studies encompassed 1910 eligible patients, with 1360 included. The population characteristics may be found in the below diagram (Figure 2).

The included study overview and findings can be found summarized in Table 2.

Subsequently, the study findings were compiled into three previously defined critical domains: patient-related (Table 3), clinical-related (disease and treatment) (Table 4), and healthcare-related (Table 5). For further definition of the critical domains it was considered that patient-related critical points would include baseline (admission for cancer care) descriptions of advanced age, comorbidities, presence sarcopenia, and/or frailty including performance status, nutritional status, body composition, and gastrointestinal or other nutrition impairing symptoms present before treatment; clinical-related (disease and treatment) would include all of the above but concerning disease characteristics, treatment induced changes and clinical outcomes; lastly, the healthcare-related domain would include descriptions of clinical care, institutional and organizational issues, such as nutritional risk screening, nutrition support, access constraints, among others deemed relevant.

### 3.1. Patient-Related Critical Points

#### 3.1.1. Advanced Age

Age was described as relating to neoadjuvant chemotherapy pathological response and lower blood counts. It is an independent risk factor that significantly impacts pathological response in patients older than 60 years old (OR = 1.840, 95% CI 1.016–3.332, *p* = 0.044) [17]. Additionally, older age was significantly associated with both a lower (*p* = 0.007) pre-chemotherapy prognostic nutritional index (PNI) [21] and a high (48.2% vs. 31.9%, *p* = 0.010) controlling nutritional status (CONUT) score [19]. Surprisingly, age did not arise as a risk factor for significant loss of skeletal muscle (*p* > 0.05) [13].

#### 3.1.2. Sarcopenia (Baseline, Pre-Treatment)

Sarcopenia accounted for adverse effects during treatment, including early termination of CT and reduced survival, but also a reduced BMI and body surface area (BSA). Sarcopenia at diagnosis was prevalent in three quarters (73.1%) of patients in the Rinninela et al. study [18]. Zhang et al. [5] identified sarcopenia before NT as a significant risk factor for treatment adverse effects during univariate analyses, and, subsequently, by multivariate logistic regression analyses (OR, 2.901; 95% CI, 1.205–6.983; *p* = 0.018), it remained an independent predictor for overall treatment-related adverse effects [5].

Regarding sarcopenic obesity, Palmela et al. [12] showed reduced OS (overall survival) (median survival 6 months [95% CI = 3.9–8.5] vs. 25 months for patients who were obese and did not have sarcopenia [95% CI = 20.2–38.2]; log-rank test *p* = 0.000). In the same study, sarcopenic obesity (100% vs. 28%; *p* = 0.004) and sarcopenia (64% vs. 28%; *p* = 0.069) were also associated with early termination of chemotherapy, with none of these patients capable of completing treatment plans. As such, the odds ratio of treatment termination was higher in patients with sarcopenia compared with patients without it (OR = 4.23; *p* = 0.050). When the authors analyzed muscle radiation attenuation, they also found the same outcomes (higher mean vs. lower, OR = 0.20; *p* = 0.040) [12]. Tan et al. [14] showed a median OS for sarcopenic patients of 569 days (IQ range: 357–1230 days) and for patients who were not sarcopenic of 1013 days (IQ range: 496–1318 days) (log-rank test, *p* = 0.04). However, they found no significant difference in overall survival in patients who experienced DLT compared with those that did not (810 days [IQ range: 323–1417] vs. 859 days [IQ range: 445–1269]; *p* = 0.665).

Looking at dose-limiting toxicity (DLT), only sarcopenia (multivariate analysis) was independently associated with DLT (odds ratio, 2.95; 95% confidence interval, 1.23–7.09; *p* = 0.015) [14]. On the contrary, Palmela et al. [12] found a non-significant trend for a DLT in patients with sarcopenia (64% vs. 39%; *p* = 0.181) and sarcopenic obesity (80% vs. 42%; *p* = 0.165), but no corresponding significant association with subsequent treatment response [12]. On multivariate analysis, the odds of treatment termination were higher in patients with sarcopenia (odds ratio = 4.23; *p* = 0.050).

Sarcopenic patients also seemed to have a lower BMI and BSA when compared with those who did not have sarcopenia [22].

Only one study assessed loss of skeletal muscle related to gender or comorbidities, such as type 1 diabetes, but did not find any significant association [13].

#### 3.1.3. BMI (Baseline, Pre-Treatment)

Baseline BMI (pre-NAC) is associated with adverse effects during treatments and overall survival (OS). Two studies showed that both underweight and overweight at baseline BMI seem significantly associated with OS and a significant risk factor for adverse effects (pre-treatment BMI < 18.5 kg/m^2;^ univariate analysis: HR = 2.015; *p* = 0.002; multivariate analysis: HR =1.456; *p* = 0.163) [19] and a BMI of 25 kg/m^2^ (*p* = 0.04) [5]. Zhou et al. [16] indicated that a lower BMI in this setting was also significantly associated with low skeletal muscle mass (*p* < 0.001) and higher nutritional risk scores, NRS 2002 (*p* < 0.001). A study by Rinninela et al. also showed a decrease in the mean of the BMI with FLOT (from 24.4 kg/m^2^ ± 3.7 to 22.6 kg/m^2^ ± 3.1; *p* < 0.0001) [18].

#### 3.1.4. Body Composition (Baseline, Pre-Treatment)

In the studies included, several associations were described between different body compositions and OS, but not all were significant. Patients with low skeletal muscle or, both, low skeletal and adipose mass had progressively shorter OS than patients with normal body composition parameters (3 year OS rates were 44.4% and 76.3%, respectively, for low skeletal muscle and adipose mass patients or for low skeletal muscle mass only vs. 88.2% for normal body composition parameters, *p* < 0.001). Low skeletal muscle mass (HR: 1.7; 95% CI: 1.2–3.7; *p* < 0.001) and low skeletal muscle and adipose mass (HR: 3.5; 95% CI: 1.5–15.2; *p* = 0.002) were independent prognostic factors of 3 year OS, namely after radical gastrectomy [16]. Other studies verified that, before NAT, the group with low visceral adipose tissue (VAT), defined as <120 cm^2^, had significantly shorter OS (*p* = 0.033), as did the group with low (<99.5 cm^2^) subcutaneous adipose tissue (SAT), after NAT (*p* = 0.032). In multivariate Cox regression analyses, low VAT before NAT (HR, 2542; 95% CI, *p* = 0.027) and low SAT after NAT (HR, 2.743; 95% CI, 1.248–6.027; *p* = 0.012) were significantly associated with low OS [5]. Moreover, patients with a marked loss of VAT (≥35.7%) during NAT had significantly shorter OS (*p* = 0.028) compared to those with no or minor (<35.7%) VAT losses. In this study, during NAT, marked loss of adiposity (as per VAT or SAT) was considered a risk factor for long-term survival. Marked (≥35.7%) VAT loss accompanied by marked SAT loss (high-risk group = NRS ≥ 3) independently predicted shorter OS (hazards ratio = 2.447; 95% confidence interval = 1.022–5.861; *p* = 0.045) [5]. However, Jin et al. [19] found no prognostic significance between the moderate or severe malnutrition group and the normal or light malnutrition group for OS at different times (pretreatment: *p* = 0.482; preoperative: *p* = 0.446; postoperative: *p* = 0.464, Kaplan–Meier with log-rank test).

There were no significant associations between different body compositions and progression free survival (PFS) or postoperative complications. Zhou et al. [16] found no significant differences in postoperative complications within 30 days among the different body composition groups, and Yamaoka et al. [13] found no association between postoperative complications and significant loss of skeletal muscle.

Different body compositions are related to disease-free survival (DFS). In the Zhang et al. [5] study, patients with low VAT before NT (<120 cm^2^) had significantly poor DFS (*p* = 0.022), similar to those with low VAT after NT (<106 cm^2^; *p* = 0.025). Multivariate analyses of DFS identified low VAT before NT (<120 cm^2^; HR, 2.50; 95% CI, 1.22; *p* = 0.012) and low VAT after NT (<106 cm^2^; HR, 2.51; 95% CI, 1.1725358; *p* = 0.018) as independent predictors for shorter DFS [5]. Moreover, patients with a marked loss of VAT (≥35.7%) during NT had significantly shorter DFS (*p* = 0.03). Simultaneously, marked VAT loss with marked SAT loss (the high-risk group) was an independent predictor for shorter DFS (hazards ratio = 2.67; 95% confidence interval = 1.182–6.047; *p* = 0.018) [15].

In most studies, there was no significant relation between body composition and tumor pathological response, except for Rinninela et al., where a decrease higher than 5% in SMI was associated with a higher Mandard tumor regression grade [5,18], whereas Jiang et al. reported that weight loss significantly influences the pathological response to treatment [17].

#### 3.1.5. Nutritional Markers and Indices

Regarding nutritional markers, patients with low skeletal muscle and adipose mass had a higher incidence of hypoalbuminemia (*p* < 0.001), lower prealbumin (*p* < 0.001), and lower IGF-1 levels (*p* = 0.031). Despite this, there were no significant differences in the preoperative concentrations of retinol-binding protein and transferrin [16]. Zhang et al. [15] found correlations between a marked loss of VAT and lower albumin levels (*p* < 0.05).

Associations between nutritional indices and OS are not consistent. Jin et al. [19] confirm that a high pre-treatment CONUT score (HR, 1.618; 95% CI, 1.111–2.356; *p* = 0.012) was independently associated with worse OS. According to Li et al. [20], PNI, albumin, and modified systemic inflammation score (mSIS) showed no significant difference after NT, and none of the pre-NT markers were independent prognostic factors for OS. However, OS was better in the pre-chemotherapy PNI-high group (3 year survival rate: 66.0% vs. 43.5%; 5 year survival rate: 55.5% vs. 25.6%, HR = 2.237, 95% CI = 1.271–3.393, *p* = 0.005), but there were no significant differences in OS between the post-chemotherapy groups (3 year survival rate: 61.5% vs. 61.9%; 5 year survival rate: 49.8% vs. 49.0%, *p* = 0.775) [21].

A high pre-treatment CONUT score (HR, 1.615; 95% CI, 1.112–2.347; *p* = 0.012) was independently associated with worse PFS [21].

Anemia and lymphocytopenia were significantly associated with a lower pre-chemotherapy PNI (*p* < 0.05) [21]. In the Sun et al. [21] study, pre-chemotherapy PNI was an independent prognostic factor (HR = 1.963, 95% CI = 1.101–3.499, *p* = 0.022), but no association was found between PNI and surgical complications (*p* = 0.157).

**Table 3 nutrients-15-02241-t003:** Patient-related critical points: summary and findings.

Study and Country	Study Design	Tumor Type, Setting, and Sample Size	Study Description	Outcomes
Jiang et al. [17] China 2021	cohort (retrospective)	Gastric adenocarcinoma; Radical surgery after NAC; *n* = 203.	Body weight recorded at two-time points: evaluated before and after NAC (before the surgery)	Weight loss was independent risk factor influencing NAC pathological responses: - >2.95% of body weight loss during NAC worsens CT response
Jin et al. [19] China 2021	cohort (retrospective)	Gastric adenocarcinoma; NAC; *n* = 272.	Serum albumin, total lymphocyte count, CONUT score. Blood samples: - within 2 weeks before the initial CT; - within 1 week before surgery; - at least 7 days after surgery (discharge)	No change in the Moderate/severe MN status during NAT Moderate/severe MN status increased postoperatively MN group: worse association with high pre-treatment CONUT score Older age associates with a high CONUT score
Sun et al. [21] China 2016	cohort (retrospective)	GC; Preoperative CT and radical surgery; *n* = 117.	Markers for the PNI score: serum albumin, total lymphocyte count. Blood samples - 1 week before NAC - within 1 week before surgery. Patients PNI-high (≥45) and PNI-low (<45).	Pre-NAC PNI not associated with surgical complications. Anemia and lymphocytopenia associates with lower pre-NAC PNI. Pre-NAC PNI is an independent prognostic factor. Higher survival for PNI-high pre-NAC patients. No differences in survival for post-CT groups. Low pre-CT PNI associates with older age.
Yamaoka et al. [13] Japan 2014	cohort (retrospective)	Primary GC; Open total gastrectomy with roux-en-y; *n* = 102 (none or adjuvant CT < 6 months) *n*= 38 (adjuvant CT > 6 months).	CT Scan - preoperatively; - postoperatively (1 year);	Loss of skeletal muscle was not associated with postoperative complications. NAC was an independent risk factor for loss of skeletal muscle. SMI decreased with NAC. Loss of skeletal muscle was not associated with sex, age, diabetes.
Zhang et al. [15] China 2021	cohort (retrospective)	GC Laparoscopic radical gastrectomy with D2 lymph node dissection followed by roux-en-y or billroth I reconstruction. NAC or CT (SOX, XELOX or FOLFOX); *n* = 110.	Skeletal muscle, VAT and SAT: - Evaluated before and after NAC (before the surgery).	Low VAT before NAC and low SAT after NAC was associated with low OS. Low VAT before and after NAC independent predictors for shorter DFS. Sarcopenia before NAC predicted adverse effects. Body composition and tumor pathological response were not significantly associated. Higher BMI after NAC was associated with postoperative complications. Higher VAT was associated with higher incidence of postoperative complications
Rinninela et al. [18] Italy 2021	cohort (retrospective)	Gastric adenocarcinoma; NAC; *n* = 26	Lumbar CTScan SMI and adipose indices: - Before FLOT - After FLOT	Almost ¾ of patients were sarcopenic at diagnosis
Zhang et al. [15] China 2021	cohort (retrospective)	Advanced GC (including gastroesophageal junction); Radical gastrectomy and NAC or CT. *n* = 157.	CTScan Skeletal muscle, VAT and SAT measure: - Before NAT - After NAT	Marked loss of VAT, marked loss of SAT predicted shorter OS and DFS. Skeletal muscle mass loss did not correlate well with nutritional status. Marked loss of VAT and lower albumin levels not related.
Palmela et al. Portugal 2017	cohort (retrospective)	Locally advanced adenocarcinoma from the stomach or gastroesophageal junction; NAC; *n* = 48.	CTScan - cancer diagnosis; - completion of NAC (*n* = 43)	Higher percentage of DLT in sarcopenic/sarcopenic obese patients (non-significant trend). Survival reduction in sarcopenic obese patients. Sarcopenic patients was associated with early CT termination (non-significant).
Zhou et al. [16] China 2020	cohort (retrospective)	GC Radical gastrectomy; *n* = 187.	Definition of gender-specific skeletal muscle/adipose cut-off values: BCS0 (normal) BCS1 (low skeletal muscle only) BCS2 (both low)	BCS2 group progressively shorter OS NAT was not the 3y OS independent prognostic factor after radical gastrectomy. BCS2 group associated with lower BMI and higher NRS2002 score. Body composition does not affect post-surgery complications. BCS2 group worse preoperative markers (hypoalbuminemia, lower prealbumin and IGF-1 levels).
Tan et al. [14] UK 2015	cohort (retrospective)	Oesophagogastric cancer; NAC; *n* = 89	Combination of CTScan, endoscopic ultrasound (EUS) and laparoscopy. Pre-treatment serum albumin levels, neutrophil-lymphocyte ratio, weight, height.	Median OS for sarcopenic patients was lower than for not sarcopenic patients. No significant difference in OS in patients who experienced DLT compared with those that did not. Sarcopenic patients had lower BMI and BSA. BMI, BSA and sarcopenia were associated with DLT.

Legend: NAC—neoadjuvant chemotherapy; LA—locally advanced; GC—gastric cancer; GEJ—gastroesophageal junction; DLT—dose-limiting toxicity; NAT—neoadjuvant treatment; CT—chemotherapy; VAT—visceral adipose tissue; SAT—subcutaneous adipose tissue; DFS—Disease free survival; MN—malnutrition; BMI—body mass index; PRNS—prognostic-related nutritional score; mSIS—modified systemic inflammation score; CT Scan—computed tomography scan; FLOT—fluorouracil plus leucovorin, oxaliplatin, and docetaxel; PGSGA—patient-generated subjected global assessment; ONS—oral nutritional supplements; CONUT—controlling nutritional status; PA—prealbumin; SMI—skeletal muscle index.

### 3.2. Clinical-Related Critical Points (Disease and Treatment)

The independent prognostic factor for 3-year OS after radical gastrectomy was tumor stage III (HR: 4.1; 95% CI: 2.1–17.8; *p* < 0.001) [16]. According to Jiang et al. [17], the independent risk factors influencing the effect of neoadjuvant chemotherapy were histological types. In the same study, clinical T stage and histological type of biopsy significantly influenced pathological response to the treatment [17].

The pathological stage was not associated with a significant loss of skeletal muscle [13]. However, Jiang et al. [17] described that those patients that did not lose weight had a better, although not significant, trend for pathological response than patients suffering from weight loss (66.4% vs. 53.3%, *p* = 0.059). Likewise, Rinninela et al. described a change in body composition (a decrease in SMI of ≥5%) and a lack of tumor-regressive changes [18].

Deep tumor invasion (*p* = 0.025) and a lower pathological complete response rate (1.2% vs. 6.6%, *p* = 0.107) were significantly associated with a higher CONUT-score [19], while Li et al. [20] found no significant difference in PNI, albumin, or mSIS after NAC.

**Table 4 nutrients-15-02241-t004:** Clinical (disease and treatment)-related critical points: summary and findings.

Study and Country	Study Design	Tumor Type, Setting, and Sample Size	Study Description	Outcomes
Zhou et al. [16] China 2020	cohort (retrospective)	G; Radical gastrectomy; *n* = 187.	Gender-specific skeletal muscle/adipose cut-off values: BCS0 (normal) BCS1 (low skeletal muscle only) BCS2 (both low)	Body composition does not affect post-surgery complications. BCS2 group worse preoperative markers (hypoalbuminemia, lower prealbumin and IGF-1). BCS2 group progressively shorter OS. NAT was not the 3y OS independent prognostic factor after radical gastrectomy.
Yamaoka et al. [13] Japan 2014	cohort (retrospective)	Total gastrectomy with roux-en-y; *n* = 102 (none or adjuvant CT < 6 months) *n*= 38 (adjuvant CT > 6 months).	CT Scan: - preoperatively; - postoperatively (1 year);	SMI decreased with NAC (independent risk factor for loss of skeletal muscle). Loss of skeletal muscle was not associated with pathological stage, preoperative SMI and ATI. Loss of skeletal muscle was not associated with postoperative complications.
Li et al. [20] China 2020	cohort (Prospective)	Gastric adenocarcinoma; Gastrectomy and NAC *n* = 225	Nutritional markers (serum albumin, BMI, PNI): - pre-NAC - post-NAC	No significant differences in PNI, Alb, and mSISo after NAT.
Zhang et al. [15] China 2021	cohort (retrospective)	GC Laparoscopic radical gastrectomy, D2 lymph node dissection Neoadjuvant CT or CT-radiotherapy (SOX, XELOX or FOLFOX); *n* = 110	Skeletal muscle, VAT and SAT; CT Scan: - before NAT - after NAT	Sarcopenia before NAT is a significant and independent predictor for overall treatment AEs; Higher BMI after NAT was significantly correlated with postoperative complications; High VAT was significantly associated with higher incidence of postoperative complications; Low VAT before NAT and low SAT after NAT was significantly associated with low OS; Low VAT before and after NAT were independent predictors for shorter DFS; No significant association between body composition and tumor pathological response.
Rinninela et al. [18] Italy 2021	cohort (retrospective)	Gastric adenocarcinoma; NAC; *n* = 26	Lumbar CTScan SMI and adipose indices: - Before FLOT - After FLOT	BMI, SMI, and VAI variations were not associated with short outcomes: - toxicity - delay and completion of perioperative FLOT - RECIST, response - the execution of gastrectomy; A decrease in SMI ≥ 5% was associated with a higher Mandard tumor-regression grade Preoperative FLOT was associated with a reduction in SMI, BMI, and VAI
Jin et al. [19] China 2021	cohort (retrospective)	Gastric adenocarcinoma; NAC; *n* = 272	Serum albumin, total lymphocyte count, CONUT score. Blood samples: - within 2 weeks before the initial CT; - within 1 week before surgery; - at least 7 days after surgery (discharge)	No change in the moderate/severe MN status during NAT. Moderate/severe MN status increased postoperatively. No association between CONUT-score and postoperative complication. CONUT-high score associates: invasion and lower pathological complete response rate. For PFS and OS: no prognostic significance between MN groups.
Jiang et al. [17] China 2021	cohort (retrospective)	Gastric adenocarcinoma; Radical surgery after NAC; *n* = 203	Body weight recorded at two-time points: - before; - after NAC (before the surgery)	Weight loss was independent risk factor influencing NAC pathological responses: - >2.95% of body weight loss during NAC worsens chemotherapy response - maintaining weight trends (non-significant) better pathological response

Legend: NAC—neoadjuvant chemotherapy; NAT—neoadjuvant treatment; CT—chemotherapy; GC—gastric cancer; AEs—adverse events; VAT—visceral adipose tissue; SAT—subcutaneous adipose tissue; SMI—skeletal muscle index; OS—overall survival; PFS—progression free survival; MN—malnutrition; BMI—body mass index; CTScan—computed tomography scan; CONUT—controlling nutritional status; PRNS—prognostic-related nutritional score; PA—prealbumin; RECIST—response evaluation criteria in solid tumors; FLOT—fluorouracil plus leucovorin, oxaliplatin, and docetaxel.

### 3.3. Healthcare-Related Critical Points

It is known that the identification of nutritional risk by assessment tools and higher scores achieved by PG-SGA are more associated with postsurgical complications, such as anastomotic leakage and intra-abdominal infection [24]. Zhao et al. [22] found that the trial group had a higher BMI than the control group (*p* < 0.005), and on the eighth day after surgery, the rate of malnutrition according to the PG-SGA and nutritional risk according to the NRS-2002 became lower in the trial group (*p* < 0.05). This group had a faster gastrointestinal recovery, a shorter-term use of drainage tubes, a shorter hospital length of stay, fewer complications (*p* < 0.05), and higher concentrations of serum prealbumin, total proteins, and albumin (*p* < 0.05) [22].

Regarding nutritional support, the group using immunonutrition intervention had fewer infectious complications when compared with the conventional intervention group, but the differences were not statistically significant (41.1% vs. 48.1%; *p* = 0.413). Although the immunonutrition group had a lower percentage of patients who were readmitted for surgical complications than the conventional group, this difference was also not significant. Claudino et al. found no significant difference in survival rates at 6 months (92.6% versus 85.0%; *p* = 0.154), 1 year (87.0% versus 78.5%; *p* = 0.153), and 5 years (69.6% versus 58.3%; *p* = 0.137). Nevertheless, the immunonutrition patient group showed a trend for longer survival when compared with the conventional nutritional group [23].

Patients without weight loss had a higher rate of oral nutritional supplements than patients with weight loss during neoadjuvant chemotherapy (82.3% vs. 70%, χ^2^ = 4.261, *p* = 0.039) [17].

**Table 5 nutrients-15-02241-t005:** Healthcare-related critical points: summary and findings.

Study and Country	Study Design	Tumor Type, Setting, and Sample Size	Study Description	Outcomes
Zhao et al. [22] China 2018	Randomized clinical trial	Adenocarcinoma of the esophagogastric junction; NAC and radiotherapy; *n* = 66	Control group: routine preoperative diet (35 kcal/kg/day) and research group: 500 mL of EN suspension ^#^ Data collected 48 h within the first hospitalization, the first day after NT and the first and eighth day after surgery	Higher BMI, serum PA, TP and ALB in trial group and a faster gastrointestinal recovery, shorter term use of drainage tubes, shorter hospital stay and less complications. Preoperative EN and ALB were independent risk factors for PRNS. Lower NRS2002 and PGSGA in the trial group
Claudino et al. [23] Brazil 2019	cohort (retrospective)	Stomach cancer; Patients who did or did not undergo NAC and who did undergo subtotal or total gastrectomy; *n* = 164.	The patients were divided into 2 groups: the immunonutrition group (received immune-modulatory diet oral or enteral, polymeric, hyperproteic diet, enriched with arginine, omega-3 fatty acids and nucleotides total 600 mL/d and 600 kcal/d for 5 to 7 days before surgery with at least 80% adherence) and conventional group	- immunonutrition group had less infectious complications compared with the conventional group, and had a lower percentage of patients who were readmitted for surgical complications than the conventional group, although differences were not significant; - immunonutrition group showed a trend for longer survival compared with the conventional nutrition group. - no significant difference in survival rates at 6 months or 1 year.
Zhao et al. [22] China 2018	Randomized clinical trial	Locally advanced gastric cancer; NAC; *n* = 106.	Patients were randomly assigned to the ^$^ ERAS or standard care group.	- serum PA, TP, and ALB concentrations were higher in the ERAS group than in the standard group.
Jiang et al. [17] China 2021	cohort (retrospective)	Gastric adenocarcinoma; Radical surgery after NAC; *n* = 203.	Body weight was recorded at the starting of NAC and before surgery, but after the last NAC. Patients with declining body weight during NAC were classified as weight loss group and patients who maintained/increased their weight during NAC were classified as no weight loss group.	Maintaining weight trends (non-significant): >higher rate of ONS usage.

Legend: NAC—neoadjuvant chemotherapy; LA—locally advanced; GC—gastric cancer; GEJ—gastroesophageal junction; DLT—dose-limiting toxicity; NAT—neoadjuvant treatment; CT—chemotherapy; VAT—visceral adipose tissue; SAT—subcutaneous adipose tissue; DFS—disease free; MN—malnutrition; BMI—body mass index; PA—prealbumin; PRNS—prognostic-related nutritional score; mSIS—modified systemic inflammation score; CT Scan—computed tomography scan; PGSGA—patient-generated subjective global assessment; ONS—oral nutritional supplements; CONUT—controlling nutritional status; SMI—skeletal muscle index. ^#^ Nutrison fiber and oral nutritional supplementation (500 mL per bottle containing 500 kcal, 20 g protein, 19.45 g fat, and 61.5 g CH); 7 days before surgery apart from routine preoperative diet (35 kcal/kg/day). Both groups on Nutrison fiber within 48 h after surgery. ^$^ ERAS group: sufficient preoperative patient education, normal diet until 6 h before surgery, liquid intake until 2 h before surgery, preoperative carbohydrate loading before surgery, analgesia with nonsteroidal anti-inflammatory drugs, minimization of opioid pain management, avoidance of perioperative fluid overload, no routine use of NGT, no abdominal drains, early removal of bladder catheters, liquid diet on recovery from anesthesia, semi-liquid diet on return of bowel function, tolerated liquid diet and forced ambulation on the day of the surgery; NGT placed preoperatively and remained until flatus occurred, intra-abdominal drains placed during surgery until the day before discharge, not allowed oral intake until bowel flatus gastrointestinal movement occurred, usually remained in bed for approximately 2 days after surgery. # Conventional group: gastrointestinal preparation before surgery, fasting from midnight, NGT placed preoperatively and remained until flatus occurred, intra-abdominal drains placed during surgery until the day before discharge, not allowed oral intake until bowel flatus gastrointestinal movement occurred, usually remained in bed for approximately 2 days after surgery.

## 4. Discussion

Gastric cancer (GC) is one of the most significant malignancies worldwide, with an annual burden prediction of ~1.8 million new cases and ~1.3 million deaths by 2040 [25]. Preoperative nutritional status is known to affect prognosis, OS, and DFS rates in surgical patients [26]. Indeed, the presence of MN in patients with radical surgical resections contributes to an increased incidence of postoperative complications and extended hospitalization [27].

It has been shown that NAC improves the overall therapeutic effects in locally advanced GC patients and does not increase the incidence of surgical complications. Additionally, undergoing GC surgery without previous NAC might significantly decrease the chance of effective reduction and radical resection [28]. NAC has been established because it confers clinical benefits over surgery [9], and it seems to be capable of enhancing immunological status, ameliorating GC patients’ postoperative prognosis. Nevertheless, these widely adopted treatment proposals (e.g., FLOT) are also known to be frequently associated with a variety of gastrointestinal adverse effects, including anorexia, nausea, vomiting, stomatitis, and diarrhea, which can lead to a further deterioration of a patient’s nutritional status, especially because these frequently present an already high risk of MN [29]. Furthermore, nutritional-related problems are one of the leading causes of hospital readmissions. Commonly, patients are not able to meet nutritional needs because of inadequate intake due to intolerance to oral and/or enteral feedings, typically manifested by nausea, vomiting, and/or early satiety [18]. For all these reasons, this review attempted to identify nutrition-related critical points during GC neoadjuvant management and their associations with clinical outcomes, as described in the selected literature.

Fourteen studies were analyzed, with 1360 patients included. Most studies were related to body composition and nutritional indexes. The results can be categorized as patient- and clinical- (disease- and treatment-) related ones. This review found considerably fewer concerning healthcare-related critical points, besides the application of nutritional risk identification tools.

Sarcopenia was predominantly considered a significant risk factor for adverse effects or the worst outcomes during treatment [5]. In addition, lower BMI and BSA relate to DLT and seem to lead to early treatment termination [14]. Interestingly, and still concerning the relationship of BSA with DLT, sarcopenic obesity was indeed associated with early treatment termination and reduced survival [12].

In these studies, GC patients’ clinical outcomes, including OS, were shown to be closely related to many nutritional parameters, such as body weight. In fact, a lower BMI was associated with a poor OS [19], while a higher BMI seems to also be a significant risk factor for adverse effects during treatments [5]. Importantly, patients who lose weight during NAC seem to be at higher risk of worse CT effects. CT adverse effects, such as nausea, vomiting, and dysgeusia, may compromise food intake, which in turn could exacerbate weight loss. This weight loss is often sharp and marked and may contribute to the loss of skeletal muscle and to MN, which might account for the description of a low BMI being related to a poorer OS. Even though BMI signifies a relationship between weight and height and cannot describe body compartments. Furthermore, NAC trajectories are long, and the timing of some of the body composition analyses might not capture the dynamic nature of the body composition variations throughout treatment.

Adding on, GC patients, who simultaneously present with a high BMI and sarcopenia, had a higher BSA but low muscle mass [14]. This is an important consideration, as it is now established that patients with low muscle mass during CT treatments will have higher toxicity and more treatment interruptions. When compared with patients with normal muscle mass, sarcopenic obesity seems capable of shaping low OS [12]. Visceral adipose tissue, strongly linked with inflammation, is shown to have a higher risk of relapse in several cancer types. Here, DFS is also associated with low VAT, both before and after NT [12]. Indeed, adiposity levels are known to be associated with both increased cancer incidence and progression in multiple tumor types, and obesity is estimated to contribute to up to 20% of cancer-related deaths [30]. Adipose tissue mechanistically disrupts physiological homeostasis, but the underlying relationships between obesity and cancer are still poorly understood.

Concerning a patient’s pathological response following treatment, this was also associated with weight loss, even though body composition did not seem to be. In addition, patients with low skeletal muscle and adipose mass had a higher incidence of hypoalbuminemia and low IGF-1 levels.

Regarding postoperative complications (within 30 days), Zhou et al. failed to show a significant association with body composition. Nonetheless, a higher BMI with a high VAT after NAC was significantly correlated with postoperative and treatment complications [15,16].

In relation to nutritional interventions, immunonutrition did not seem to have a significant association with complications or survival rates. On the other hand, patients with nutritional support strategies, such as oral nutritional supplements, were shown to have better weight stability throughout the proposed treatments [22,23].

Age has been found to be associated with physiological changes influencing drug pharmacokinetics, thus affecting cancer therapies [31]. In this review, one study related age to pathological response [17], showing a better pathological response in older patients than in younger ones. This could imply a more aggressive gastric cancer in younger patients and, hence, a poorer clinical response. Interestingly, older patients present lower CONUT and PNI scores, indicative of lower serum albumin and lymphocyte counts. Ageing also carries the risk of an impaired immune and hematologic system, potentially making elderly patients more vulnerable to infections and, in turn, more susceptible to earlier treatment termination [31].

Yamaoka et al. found that age was not a risk factor associated with a significant loss of skeletal muscle after total gastrectomy, even though it is expected that a higher percentage of muscle wasting occurs in the elderly over 65 years of age [13].

This systematic review tried to clarify the exposures and critical determinants that may be impacting GC patients’ nutritional status during neoadjunvancy, and our findings seem to reinforce the importance of body composition throughout the course of NT. GC is known to be accompanied by MN, altered metabolism, and cancer-associated cachexia, with a significant impact on the patient’s nutritional status, muscle compartments, function, and OS [32]. GC patients will then be exposed to the burden of persistent inflammation and metabolic deregulation, along with decreased food intake due to anorexia, nausea, and digestive impairments such as epigastric pain and early satiety. Many of these symptoms endured since clinical presentation and/or diagnosis, if unabated, will potentially be made worse by the prolonged multimodal treatment, which, in turn, might aggravate any involuntary weight loss or sarcopenia [26,33]. Although current guidelines already recommend screening and the systematic identification of nutritional risk as the first step for the nutritional care process of cancer patients, and as sarcopenia’s independent prognostic value becomes more established, body composition assessment could emerge as a broader tool to support clinical decision making in patients with GC, namely dose and toxicity management [34].

Regardless, the exact role of nutritional support during NAC has yet to be fully explored. Even though the evidence shows that nutritional support in the immediate perioperative period with immune-nutrient-enriched formulas seems to reduce surgical complications, little is known about the type of nutritional interventions during NAC [26,33].

In addition, it is urgent to better comprehend the role of nutritional support in stabilizing and reversing sarcopenia and its role during cancer-associated body composition changes, specifically throughout NAC.

Most of the studies found and included in this review had a retrospective design and recruited a small sample size (single center). Many have also identified the following limitations: heterogeneous clinical data, inconsistencies in the prescribed treatment plan, time to follow up, and diverse cut-off values (Appendix A). This review has also identified a lack of studies documenting wider aspects that might influence nutritional status, such as healthcare and organizational critical points. Patients with NT proposals will be exposed for longer to treatments and hospital visits, and this might be even more concerning for those having to accommodate farther travel to reach reference centers. More is needed to better understand the nutritional status implications of these prolonged care continuum exposures and subsequent clinical outcomes.

Finally, NT is a period that normally encompasses several weeks and could undoubtedly represent an opportunity to identify, manage, and tackle nutritional-related issues that seem to be associated with several clinical outcomes and to provide the best supportive measures for GC patients.

## 5. Conclusions

Neoadjuvant chemotherapy in gastric cancer patients has the potential to contribute to an increase in catabolic stress, nutritional impact symptoms, malnutrition, and sarcopenia. Pursuing a better understanding of the exposure to critical domains affecting nutritional status risk and their determinants will enable proactive clinical approaches and optimized care plans by deploying appropriate and timely nutrition support so that there is an opportunity to mitigate poor nutritional status and sarcopenia alongside their deleterious clinical consequences.

## Figures and Tables

**Figure 1 nutrients-15-02241-f001:**
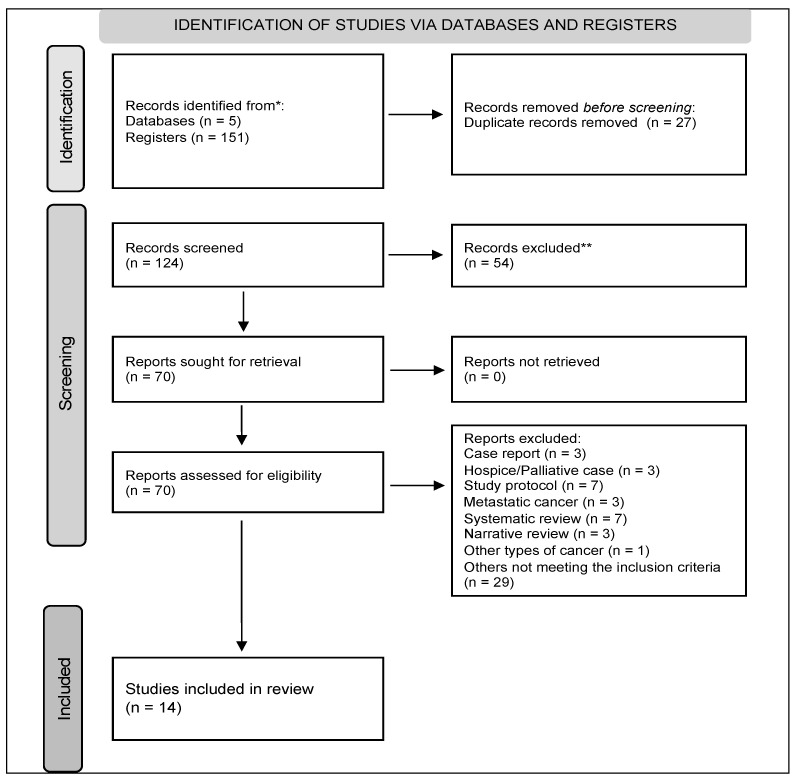
Flow chart of studies’ selection (PRISMA) [11]. * Pubmed/Medline, US National Library of Medicine’s PubMed, ISI’s Web of Knowledge, Cochrane, and Scopus databases; ** records that were excluded from analysis.

**Figure 2 nutrients-15-02241-f002:**
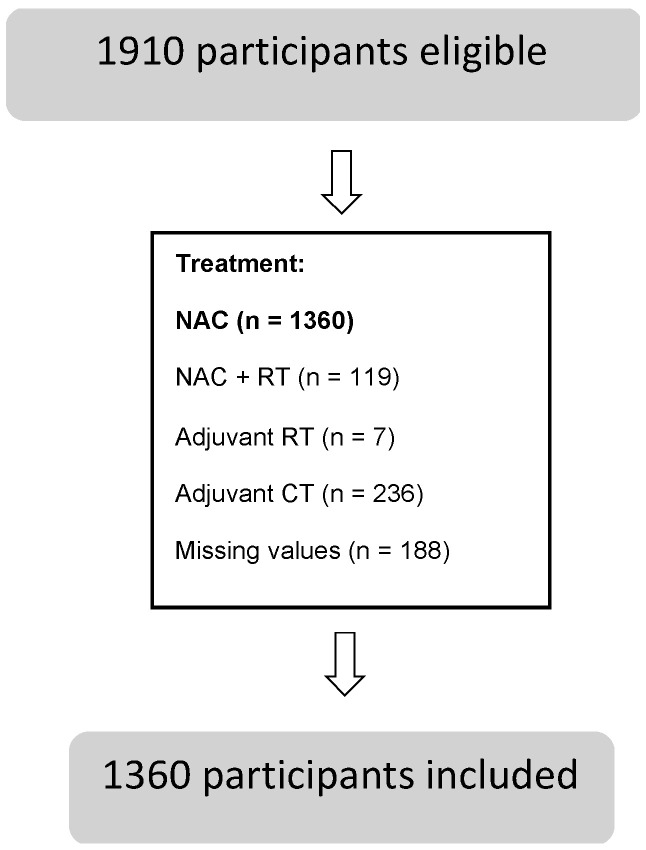
Population characteristics. Legend: NAC—neoadjuvant chemotherapy; CT—chemotherapy; RT—radiotherapy.

**Table 1 nutrients-15-02241-t001:** Inclusion and exclusion criteria.

Criteria	Inclusion	Exclusion
Patients’ characteristics	Human adults aged ≥ 18 years	≤18 years, pregnant women
	Medical oncology outpatients	Patients hospitalized: wards, care in acute or intensive or critical or long-term or end of life units. Surgical patients. Palliative patients.
Disease characteristics	Histologically documented primary gastric cancer suitable for a neoadjuvant treatment approach: - locally advanced gastric cancer, - newly diagnosed - without any prior antitumor treatment, - potentially resectable disease - clinically diagnosed stage: cT2-4/cN-any/cM0 or according to reported ultrasound, endoscopy, or enhanced CT/MRI scan: cT any/cN +/cM0.	Healthy In situ disease Other early stages Metastatic settings
Outcomes	Nutrition-critical domains: Patient-related critical points Clinical-related critical points (disease and treatment) Healthcare-related critical points	
Language	English, Portuguese, Spanish, and French	
Year	2011–2021	All other years
In cases of uncertainties about the data reported, the trials’ authors are contacted in order to obtain more information; if contact is not possible, a team consensus decision is made about the inclusion or exclusion of studies.		

**Table 2 nutrients-15-02241-t002:** Overview of studies and summary of findings.

Study, Country Year	Study Design	Tumor Type, Setting, and Sample Size	Study Description	Outcomes		
				Clinical Characteristics (OS, DFS, PFS, Age Comorbidities)	Treatment Complications (DLT, Completion)	Surgery-Related Events
Body Composition Studies						
Palmela et al. [12] Portugal 2017	cohort (retrospective)	Locally advanced (LA) gastric or (GEJ) adenocarcinoma; NAC; *n* = 48.	CT Scan - cancer diagnosis; - completion of NAC (*n* = 43)	Survival reduction in sarcopenic obese patients. (median survival 6 months [95% CI = 3.9–8.5] vs. 25 months for patients who were obese and did not have sarcopenia [95% CI = 20.2–38.2]; log-rank test *p* = 0.000)	Higher percentage of DLT in sarcopenic/sarcopenic obese patients (non-significant trend). DLT in patients with sarcopenia (64% vs. 39%; *p* = 0.181) and sarcopenic obesity (80% vs. 42%; *p* = 0.165 Sarcopenic patients was associated with early CT termination (non-significant). (sarcopenic obesity (100% vs. 28%; *p* = 0.004) and sarcopenia (64% vs. 28%; *p* = 0.069) associated with early termination of CT; OR = 4.23; *p* = 0.050)	
Yamaoka et al. [13] Japan 2014	cohort (retrospective)	Gastric cancer; Open total gastrectomy (roux-en-y) - None or adjuvant CT < 6 months *n* = 102 - Adjuvant CT > 6 months *n*= 38	CT Scan - preoperatively; - postoperatively (1 year);		SMI decreased with NAC. Loss of skeletal muscle was not associated with sex, age (*p* > 0.05), diabetes, pathological stage, and preoperative SMI and ATI.	Loss of skeletal muscle was not associated with postoperative complications. NAC was an independent risk factor for loss of skeletal muscle.
Tan et al. [14] UK 2015	cohort (retrospective)	Esophagogastric cancer; NAC; *n* = 89	Combination of CT Scan, endoscopic ultrasound (EUS) and laparoscopy. Pre-treatment serum albumin levels and neutrophil-lymphocyte ratio, weight and height.	Median OS for sarcopenic patients was lower than for not sarcopenic patients. (569 days (IQ range: 357–1230 days) and for not sarcopenic 1013 days (IQ range: 496–1318 days, log-rank test, *p* = 0.04) No significant difference in OS in patients who experienced DLT compared with those that did not. (810 days [IQ range: 323–1417] vs. 859 days [IQ range: 445–1269]; *p* = 0.665)	Sarcopenic patients had lower BMI and BSA. BMI, BSA and sarcopenia were associated with DLT. (OR 2.95; 95% confidence interval, 1.23–7.09; *p* = 0.015)	
Zhang et al. [15] China 2021	cohort (retrospective)	Gastric cancer; Laparoscopic radical gastrectomy D2 lymph node dissection NAC; *n* = 110.	CT Scan Skeletal muscle, VAT and SAT: - Before NAC - After NAC (before the surgery).	Low VAT before NAC and low SAT after NAC was associated with low OS. Low VAT before and after NAC independent predictors for shorter DFS. (OR, 2.901; 95% CI, 1.205–6.983; *p* = 0.018)	Sarcopenia before NAC predicted adverse effects. Body composition and tumor pathological response were not significantly associated.	Higher BMI after NAC was associated with postoperative complications. Higher VAT was associated with higher incidence of postoperative complications.
Zhou et al. [16] China 2020	cohort (retrospective)	Gastric cancer; Radical gastrectomy; *n* = 187	Definition of gender-specific skeletal muscle/adipose cut-off values (CT Scan): BCS0 (normal) BCS1 (low skeletal muscle only) BCS2 (both low)	BCS2 group progressively shorter OS NAT was not the 3y OS independent prognostic factor after radical gastrectomy. (BCS2 HR: 3.5; 95% CI: 1.5–15.2; *p* = 0.002) were independent prognostic factor of 3 year OS; also low VAT before NAT (HR, 2.542; 95% CI; *p* = 0.027) and low SAT after NAT (HR, 2.743; 95% CI, 1.248–6.027; *p* = 0.012) were significantly associated with low OS	BCS2 group associated with lower BMI and higher NRS2002 score. (*p* < 0.001)	Body composition does not affect post-surgery complications. BCS2 group worse preoperative markers (hypoalbuminemia (*p* < 0.001), lower prealbumin, (*p* < 0.001), and IGF-1 levels (*p* = 0.031).
Zhang et al. [15] China 2021	cohort (retrospective)	Advanced GC Radical gastrectomy and NAC; *n* = 157	Skeletal muscle, VAT and SAT (CT Scan): - Before NAC - After NAC (before the surgery).	Marked loss of VAT, marked loss of SAT predicted shorter OS (*p* = 0.022) and DFS (Independent predictor for shorter DFS (hazards ratio = 2.67; 95% confidence interval = 1.182–6.047; *p* = 0.018) Skeletal muscle mass loss did not correlate well with nutritional status.	Marked loss of VAT and lower albumin levels not related.	
Jiang et al. [17] China 2021	cohort (retrospective)	Gastric adenocarcinoma; Radical surgery after NAC; *n* = 203	Body weight recorded at two-time points: - Before NAC - After NAC (before the surgery).	Independent risk factor for pathological response: - age (OR = 1.840, 95% CI 1.016–3.332, *p* = 0.044) - histological type	Weight loss was independent risk factor influencing NAC pathological responses: - >2.95% of body weight loss during NAC worsens CT response - maintaining weight trends (non-significant, (66.4% vs. 53.3%, *p* = 0.059): - better pathological response - higher rate of ONS usage Patients without weight loss had a higher rate of oral nutritional supplements than patients with weight loss during NAC (82.3%% vs. 70%, χ^2^ = 4.261, *p* = 0.039)	
Rinninela et al. [18] Italy 2021	cohort (retrospective)	Gastric adenocarcinoma; NAC; N = 26	CTScan Preoperative pre- and post-FLOT Lumbar SMI and adipose indices: - Before FLOT - After FLOT		BMI, SMI, and VAI variations not associated with short term outcomes: - Toxicity, delay and completion of perioperative FLOT (BMI from 24.4 kg/m^2^ ± 3.7 to 22.6 kg/m^2^ ± 3.1; *p* < 0.0001) - RECIST and Mandard A decrease of SMI ≥ 5% associates with a higher Mandard tumor-regression grade	Execution of gastrectomy not related with BMI, SMI, and VAI variations.
Nutritional markers						
Jin et al. [19] China 2021	cohort (retrospective)	Gastric adenocarcinoma; NAC; *n* = 272	Serum albumin, total lymphocyte count, CONUT score. Blood samples: - within 2 weeks before the initial CT; - within 1 week before surgery; - at least 7 days after surgery (discharge)	For PFS and OS: - No prognostic significance between groups moderate/severe MN vs. normal/light MN group (pretreatment: *p* = 0.482, preoperative: *p* = 0.446; postoperative: *p* = 0.464, Kaplan–Meier with log-rank test) - worse association with high pre-treatment Hight pre-treatment CONUT score (HR, 1.618; 95% CI, 1.111–2.356; *p* = 0.012 independently associated with worse OS). Age Older age associates with high CONUT score (48.2% vs. 31.9%, *p* = 0.010) CONUT score) OS was better in pre-CT PNI-high group (3 year survival rate: 66.0% vs. 43.5%; 5 year survival rate: 55.5% vs. 25.6%, HR = 2.237, 95% CI = 1.271–3.393, *p* = 0.005), but there were no significant differences in OS between the post-CT groups (3 year survival rate: 61.5% vs. 61.9%, 5 year survival rate: 49.8% vs. 49.0%, *p* = 0.775)	CONUT-high-score associates, invasion, and lower pathological complete response rate. (HR, 1.615; 95% CI, 1.112–2.347; *p* = 0.012)	No change in the Moderate/severe MN status during NAT. Moderate/severe MN status increased postoperatively. No association between CONUT-score and postoperative complication.
Li et al. [20] China 2020	Cohort (prospective)	Gastric adenocarcinoma; Gastrectomy and NAC; *n* = 225	Nutritional markers (serum abumin, BMI, PNI) - pre-NAC - post-NAC		No significant differences in PNI, Alb, and mSISo after NAT (*p* > 0.05)	
Sun et al. [21] China 2016	cohort (retrospective)	Gastric cancer; NAC and radical surgery; *n* = 117	Markers for the PNI score: serum albumin, total lymphocyte count. Blood samples - 1 week before NAC - within 1 week before surgery. Patients PNI-high (≥45) and PNI-low (<45).	OS Higher OS for PNI-high pre-NAC patients; No differences in OS for post-CT groups; Age Low pre-CT PNI associates with older age (*p* = 0.007 pre-CT PNI)	Anemia and lymphocytopenia associates with lower pre-NAC PNI (HR = 1.963, 95% CI = 1.101–3.499, *p* = 0.022), Pre-NAC PNI is an independent prognostic factor.	Pre-NAC PNI not associated with surgical complications (*p* = 0.157).
Nutritional support studies						
Zhao et al. [22] China 2018	Randomized clinical trial	Adenocarcinoma of the esophagogastric junction; NAC and radiotherapy; *n* = 66	Control group: routine preoperative diet (35 kcal/kg/day) and research group: 500 mL of EN suspension # Data collected 48 h within the first hospitalization, the first day after NT and the first and 8th day after surgery		Higher BMI, serum PA, TP and ALB in trial group and a faster gastrointestinal recovery, shorter term use of drainage tubes, shorter hospital stay and less complications (*p* < 0.05)	Preoperative EN and ALB were independent risk factors for PRNS. (*p* < 0.05) Lower NRS2002 and PGSGA in the trial group (*p* < 0.05).
Claudino et al. [23] Brazil 2019	cohort (retrospective)	Gastric cancer. Subtotal or total gastrectomy. Patients who did or did not undergo NAC *n* = 164	Two groups: - immunonutrition ^¥^ before surgery - conventional	No significant difference in OS rates at 6 months, 1 year, and 5 years (no significant difference in OS rates at 6 months (92.6% versus 85.0%; *p* = 0.154) 1 year (87.0% versus 78.5%; *p* = 0.153) and 5 years (69.6% versus 58.3%; *p* = 0.137). A trend for longer OS was found in immunonutrition group.	Immunonutrition group with less infectious complications (non-significant)	Immunonutrition group with less readmissions for surgical complications (non- significant) (41.1% vs. 48.1%; *p* = 0.413)
Zhao et al. [22] China 2018	Randomized clinical trial	Locally advanced gastric cancer; NAC; *n* = 106	^$^ ERAS group or standard care group.	Sarcopenic patients had lower OS than non-sarcopenic patients (*p* < 0.05). No significant differences in OS for patients who experienced DLT.	BMI and BSA were lower in sarcopenic patients and associated with DLT.	

Legend: NAC—neoadjuvant chemotherapy; LA—locally advanced; GEJ—gastroeosophageal junction; DLT—dose-limiting toxicity; NAT—neoadjuvant treatment; CT—chemotherapy; VAT—visceral adipose tissue; SAT—subcutaneous adipose tissue; DFS—disease-free survival; MN—malnutrition; BMI—body mass index; PA—prealbumin; PRNS—prognostic-related nutritional score; mSIS—modified systemic inflammation score; CT Scan—computed tomography scan; RECIST—response evaluation criteria in solid tumors; FLOT—fluorouracil plus leucovorin, oxaliplatin, and docetaxel; PGSGA—patient generated subjected global assessment; SMI—skeletal muscle index; CONUT—controlling nutritional status. # Nutrison fiber and oral nutritional supplementation (500 mL per bottle containing 500 kcal, 20 g protein, 19.45 g fat, and 61.5 g CH); 7 days before surgery apart from routine preoperative diet (35 kcal/kg/day). Both groups on Nutrison fiber within 48 h after surgery. ¥ Immune enteral diet enriched with arginine, omega-3 fatty acids, and nucleotides. ^$^ ERAS group: sufficient preoperative patient education, normal diet until 6 h before surgery, liquid intake until 2 h before surgery, preoperative carbohydrate loading before surgery, analgesia with nonsteroidal anti-inflammatory drugs, minimization of opioid pain management, avoidance of perioperative fluid overload, no routine use of NGT, no abdominal drains, early removal of bladder catheters, liquid diet on recovery from anesthesia, semi-liquid diet on return of bowel function, tolerated liquid diet and forced ambulation on the day of the surgery; NGT placed preoperatively and remained until flatus occurred, intra-abdominal drains placed during surgery until the day before discharge, not allowed oral intake until bowel flatus gastrointestinal movement occurred, usually remained in bed for approximately 2 days after surgery. Conventional group: gastrointestinal preparation before surgery, fasting from midnight.

## Data Availability

Data is unavailable due to privacy or ethical restrictions.

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
