# Peer review of "Neoadjuvant Gastric Cancer Treatment and Associated Nutritional Critical Domains for the Optimization of Care Pathways: A Systematic Review"

_nutrients, 2023, doi:10.3390/nu15102241_

Round 1

Reviewer 1 Report

The submitted paper provides a comprehensive review of the nutritional-related critical points during gastric cancer neoadjuvant management and their associations with clinical outcomes. The authors have analyzed 14 studies with a total of 1360 patients, highlighting the importance of nutritional status, body composition, and other factors in determining clinical outcomes in gastric cancer patients. Overall, the manuscript is well-organized and provides valuable information. However, there are several major and minor points that need to be addressed to improve the overall quality and clarity of the paper.

Major points:

1) The authors should provide a clear and concise description of the search strategy and inclusion/exclusion criteria used in selecting the studies for this review. This would allow readers to better understand the methodology and assess the quality of the review.

2) In some instances, the statistical values and terminology used in the paper are unclear or inconsistent. The authors should ensure that all reported p-values, hazard ratios, and confidence intervals are clearly presented and explained throughout the text.

3) The paper would benefit from a more thorough discussion of the limitations of the included studies. For example, the authors should discuss potential biases and confounding factors that may have influenced the results of the individual studies, as well as the overall conclusions of the review.

4) It would be helpful to elaborate on the potential mechanisms underlying the associations between nutritional status, body composition, and clinical outcomes in gastric cancer patients. This would provide a more comprehensive understanding of the topic and guide future research in this area.

5) The authors should discuss the clinical implications of their findings, including any potential recommendations for the assessment and management of nutritional status in gastric cancer patients undergoing neoadjuvant treatment.

6) The conclusion section should be revised to provide a more focused summary of the main findings, as well as to address the implications for clinical practice and future research.

Minor points:

1) The manuscript contains several grammatical and typographical errors, which should be corrected for better readability and clarity.

2) Ensure consistency in the use of abbreviations throughout the manuscript, such as neoadjuvant chemotherapy (NAC) and overall survival (OS).

3) In several instances, the paper uses an excessive amount of direct quotes from the reviewed studies. Consider paraphrasing these passages and summarizing the findings in the authors' own words to improve readability and reduce redundancy.

4) The formatting of the references should be checked to ensure they adhere to the journal's guidelines.

Overall, the submitted paper provides valuable insights into the nutritional-related critical points in gastric cancer neoadjuvant management and their associations with clinical outcomes. Addressing the major and minor points outlined above will improve the quality, clarity, and impact of the manuscript.

Author Response

Major points:

  • The authors should provide a clear and concise description of the search strategy and inclusion/exclusion criteria used in selecting the studies for this review. This would allow readers to better understand the methodology and assess the quality of the review.

Thank you for the question. We also agree that it increases clarity to the manuscript the addition of the required information.

Our search strategy is detailed in the PROSPERO document and was added to the methodology of the revision.

Briefly, it consisted of: “The search strategy in this revision used the following key words (((Gastric OR Stomach) AND (Cancer OR Neoplasm OR Carcinoma OR Malignancy)) AND (Neoadjuvant OR Pre-operatory) AND (Nutritional status OR Nutritional intervention OR Nutritional support OR Dietary counselling OR Oral nutritional supplements)); and was conducted using five databases (Pubmed/medline; US National Library of Medicine's PubMed; ISI's Web of Knowledge; Cochrane database; Scopus) over a period of ten years, since the revision process started (from 2011 to 2021)”.

According to your suggestion we also now included one example of our search strategy that might fit as a supplementary data. (Appendix 1 - Supplementary Data). These changes can be found: manus line 79. Regarding better clarification of the inclusion and exclusion criteria we also agree with you and for this purpose we have now highlighted those in a new summary table (Table 1): manus line 85.

Appendix 1 - Supplementary data: search strategy example

Cochrane Library search strategy (3rd November 2021)

1#        MeSH descriptor: [Stomach Neoplasms] 3 tree(s) exploded

#          MeSH descriptor: [Neoadjuvant Therapy] this term only

#3        MeSH descriptor: [Preoperative Period] 1 tree(s) exploded

#4        #2 or #3

#5        MeSH descriptor: [Outpatients] 1 tree(s) exploded

#6        MeSH descriptor: [Ambulatory Care] explode all trees

#7        "outpatients":ti,ab,kw  (Word variations have been searched)

#8        #5 or #6 or #7

#9        "nutrition* support" ti,ab,kw  (Word variations have been searched)

#10     "oral nutrition* supplement*"

#11     "protein supplement*"

#12     "energy supplement*"

#13     "nutritional counseling"

#14     "dietary advice"

#15     "food fortification"

#16     "food enrichment"

#17     MeSH descriptor: [Nutritional Status] explode all trees

#18     MeSH descriptor: [Anthropometry] explode all trees

#19     MeSH descriptor: [Body Composition] explode all trees

#20     #9 or #10 or #11 or #12 or #13 or #14 or #15 or #16 or #17 or #18 or #19

            #21     #1 and #4 and #8 and #20

#22     MeSH descriptor: [Letter] explode all trees

#23     MeSH descriptor: [Editorial] explode all trees

#24     MeSH descriptor: [Animals] explode all trees

#25     MeSH descriptor: [Child] explode all trees

#26     MeSH descriptor: [Pediatrics] explode all trees

#27     MeSH descriptor: [Pregnant Women] explode all trees

#28     MeSH descriptor: [Inpatients] focus

#29     MeSH descriptor: [Hospital] explode all trees

#30     "ward":ti,ab,kw  (Word variations have been searched)

#31     MeSH descriptor: [Terminal Care] 2 tree(s) exploded

#32     MeSH descriptor: [Palliative Care] 2 tree(s) exploded

#33     letter or editorial or "case report"  or animal or rodent* or child* or infant or infancy or p?diatric* or pregnant or ambulatory or inpatient* or "nursing home*" or "long term care" or palliative or "end of life" or "terminal care"

#34     #22 or #23 or #24 #25 #26 or #27 or #28 or #29 or #30 or #31 or #32 or #33

#35     #21 not #34

Table 1 Inclusion and exclusion criteria.

Criteria

Inclusion

Exclusion

Patients characteristics

Human adults aged ≥18 years

≤18 years, pregnant women

Medical oncology outpatients

Patients hospitalized: wards, care in acute or intensive or critical or long-term or end of life units.

Surgical patients.

Palliative patients.

Disease characteristics

Histologically documented primary gastric cancer suitable for a neoadjuvant treatment approach:

-        locally advanced gastric cancer,

-        newly diagnosed

-        without any prior antitumor treatment,

-        potentially resectable disease

-        clinically diagnosed stage: cT2-4/cN-any/cM0 or according to reported ultrasound, endoscopy or enhanced CT/MRI scan: cT any/cN+/cM0.

Healthy

In situ disease

Other early stages

Metastatic settings

Outcomes

Nutrition critical domains:

Patient-related critical points

Clinical-related critical points (disease and treatment)

Healthcare-related critical points

Language

English, Portuguese, Spanish and French

Year

2011-2021

All other years

In case of uncertainties about the data reported, the trials' authors are contacted in order to get more information; if contact was not possible, a team consensus decision was made about the inclusion or exclusion of studies.

2) In some instances, the statistical values and terminology used in the paper are unclear or inconsistent. The authors should ensure that all reported p-values, hazard ratios, and confidence intervals are clearly presented and explained throughout the text.

Thank you for your comment. We agree with your concerns, and for clarity’s sake we extract the results from the text and incorporate them in former Table1, now Table 2.

Table 2 – Overview of studies and summary of findings

Study, Country Year

Study design

Tumour type, setting and sample size

Study description

Outcomes

Clinical Characteristics

(OS, DFS, PFS, age comorbidities)

Treatment Complications

(DLT, completion)

Surgery-related events

Body Composition Studies

Palmela et al

Portugal

2017

cohort

(retrospective)

Locally advanced (LA) gastric or (GEJ) adenocarcinoma;

NAC; n=48.

CT Scan

- cancer diagnosis;

- completion of NAC (n= 43)

Survival reduction in sarcopenic obese patients.

(median survival 6 months [95% CI=3.9–8.5] vs. 25 months for patients who were obese and did not have sarcopenia [95% CI=20.2–38.2]; log-rank test P=0.000)

Higher percentage of DLT in sarcopenic/sarcopenic obese patients (non-significant trend). DLT in patients with sarcopenia (64% vs. 39%; p=0.181) and sarcopenic obesity (80% vs. 42%; p=0.165

Sarcopenic patients was associated with early CT termination (non-significant).

(sarcopenic obesity (100% vs. 28%; p=0.004) and sarcopenia (64% vs. 28%; p=0.069) associated with early termination of CT; OR=4.23; p=0.050)

----

----

Yamaoka et al

Japan

2014

cohort

(retrospective)

Gastric cancer;

Open total gastrectomy

(roux-en-y)

-None or adjuvant CT<6 mths

n=102

-Adjuvant CT>6 mths

 n= 38

CT Scan 

- preoperatively;

- postoperatively (1 year);

-

SMI decreased with NAC.

Loss of skeletal muscle was not associated with sex, age (p>0.05), diabetes, pathological stage, and preoperative SMI and ATI.

Loss of skeletal muscle was not associated with postoperative complications.

NAC was an independent risk factor for loss of skeletal muscle.

Tan et al

UK

2015

cohort

(retrospective)

Esophagogastric cancer;

NAC;

n= 89

Combination of CT Scan, endoscopic ultrasound (EUS) and laparoscopy.

Pre-treatment serum albumin levels and neutrophil-lymphocyte ratio, weight and height.

Median OS for sarcopenic patients was lower than for not sarcopenic patients. (569 days (IQ range: 357-1230 days) and for not sarcopenic 1013 days (IQ range: 496-1318 days, log-rank test, p=0.04)

No significant difference in OS in patients who experienced DLT compared with those that did not.  (810 days [IQ range: 323-1417] vs. 859 days [IQ range: 445e1269]; p=0.665)

Sarcopenic patients had lower BMI and BSA.

BMI, BSA and sarcopenia were associated with DLT.

(OR 2.95; 95% confidence interval, 1.23-7.09; p= 0.015)

-

Zhang et al

China

2021

cohort

(retrospective)

Gastric cancer;

Laparoscopic radical gastrectomy

D2 lymph node dissection

NAC;

 n=110.

CT Scan 

Skeletal muscle, VAT and SAT:

- Before NAC

- After NAC (before the surgery).  

Low VAT before NAC and low SAT after NAC was associated with low OS.

Low VAT before and after NAC independent predictors for shorter DFS.

(OR, 2.901; 95% CI, 1.2056.983; p=0.018)

Sarcopenia before NAC predicted adverse effects.

Body composition and tumour pathological response were not significantly associated.

Higher BMI after NAC was associated with postoperative complications.

Higher VAT was associated with higher incidence of postoperative complications.

Zhou et al

China

2020

cohort

(retrospective)

Gastric cancer;

Radical gastrectomy;

n=187

Definition of gender-specific skeletal muscle/ adipose cut-off values (CT Scan):

BCS0 (normal)

BCS1 (low skeletal muscle only)

BCS2 (both low)

BCS2 group progressively shorter OS

NAT was not the 3y OS independent prognostic factor after radical gastrectomy.

(BCS2 HR: 3.5; 95% CI: 1.5-15.2; p=0.002) were independent prognostic factor of 3-year OS; alsolow VAT before NAT (HR, 2,542; 95% CI; p = 0.027) and low SAT after NAT (HR, 2.743; 95% CI, 1.2486.027; p = 0.012) were significantly associated with low OS

BCS2 group associated with lower BMI and higher NRS2002 score. (p < 0.001)

Body composition does not affect post-surgery complications.

BCS2 group worse preoperative markers (hypoalbuminemia (p < 0.001), lower prealbumin, (p < 0.001),  and IGF-1 levels (p= 0.031).

Zhang et al

China

2021

cohort

(retrospective)

Advanced gastric cancer;

Radical gastrectomy and NAC;

n=157

Skeletal muscle, VAT and SAT (CT Scan):

- Before NAC

- After NAC (before the surgery). 

Marked loss of VAT, marked loss of SAT predicted shorter OS (p=0.022) and DFS (Independent predictor for shorter DFS (hazards ratio = 2.67; 95% confidence interval = 1.182–6.047; p=0.018)

Skeletal muscle mass loss did not correlate well with nutritional status.

Marked loss of VAT and lower albumin levels not related.

-

Jiang et al

China

2021

cohort

(retrospective)

Gastric adenocarcinoma;

Radical surgery after NAC;

n=203

Body weight recorded at two-time points:

- Before NAC

- After NAC (before the surgery). 

Independent risk factor for pathological response:

- age (OR = 1.840, 95% CI 1.016–3.332, P = 0.044)  

- histological type

Weight loss was independent risk factor influencing NAC pathological responses:

- >2.95% of body weight loss during NAC worsens CT response

-maintaining weight trends (non-significant, (66.4% vs 53.3%, p=0.059):

    - better pathological response

    - higher rate of ONS usage

Patients without weight loss had a higher rate of oral nutritional supplements than patients with weight loss during NAC (82.3%% vs 70%, χ2 = 4.261, p=0.039)

-

Rinninela et al

Italy

2021

cohort

(retrospective)

Gastric adenocarcinoma;

NAC;

N=26

CTScan

Preoperative pre- and post-FLOT Lumbar SMI and adipose indices:

- Before FLOT

- After FLOT

-

BMI, SMI, and VAI variations not associated with short term outcomes:

-Toxicity, delay and completion of perioperative FLOT (BMI from 24.4 kg/m2 ± 3.7 to 22.6 kg/m2 ± 3.1; p<0.0001)

- RECIST and Mandard 

A decreases of SMI >5% associates with a higher Mandard tumour regression grade

Execution of gastrectomy not related with BMI, SMI, and VAI variations.

Nutritional markers

Jin et al

China

2021

cohort

(retrospective)

Gastric adenocarcinoma;

NAC;

n=272

Serum albumin, total lymphocyte count, CONUT score.

Blood samples:

-within 2-weeks before the initial CT;

-within 1-week before surgery;

-at least 7-days after surgery (discharge)

For PFS and OS:

-No prognostic significance between groups moderate/severe MN vs. normal/light MN group (pretreatment: p=0.482, preoperative: p=0.446; postoperative: p=0.464, Kaplan-Meier with log-rank test)

-worse association with high pre-treatment Hight pre-treatment CONUT score (HR, 1.618; 95% CI, 1.111–2.356; p=0.012 independently associated with worse OS).

Age

Older age associates with high CONUT score (48.2% vs 31.9%, p=0.010) CONUT score)

OS was better in pre-CT PNI-high group (3-year survival rate: 66.0% vs 43.5%; 5 year survival rate: 55.5% vs 25.6%, HR=2.237, 95% CI=1.271-3.393, p=0.005), but there was no significant differences in OS between the post-CT groups (3-year survival rate: 61.5% vs 61.9%, 5 year survival rate: 49.8% vs 49.0%, p=0.775)

CONUT-high-score associates, invasion, and lower pathological complete response rate.

(HR, 1.615; 95% CI, 1.112–2.347; p=0.012)

No change in the Moderate/severe MN status during NAT.

 Moderate/severe MN status increased postoperatively.

No association between CONUT-score and postoperative complication.

Li et al

China

2020

Cohort

(prospective)

Gastric adenocarcinoma;

Gastrectomy and NAC;

n=225

Nutritional markers (serum abumin, BMI, PNI)

-pre-NAC

-post-NAC

-

No significant differences in PNI, Alb, and mSISo after NAT (p>0.05)

Sun et al

China

2016

cohort

(retrospective)

 Gastric cancer;

NAC and radical surgery;

n=117

Markers for the PNI score: serum albumin, total lymphocyte count.

Blood samples

-1-week before NAC

-within 1-week before surgery.

Patients PNI-high (≥45) and PNI-low (<45).

OS

Higher OSfor PNI-high pre-NAC patients;

No differences in OS for      post-CT groups;

Age

Low pre-CT PNI associates with older age (p=0.007 pre-CT PNI)

Anaemia and lymphocytopenia associates with lower pre-NAC PNI(HR=1.963, 95% CI=1.101-3.499, p=0.022),

Pre-NAC PNI is an independent prognostic factor.

Pre-NAC PNI not associated with surgical complications (p=0.157).

Nutritional support studies

Zhao et al

China

2018

Randomised clinical trial

Adenocarcinoma of the esophagogastric junction;

NAC and radiotherapy;

n=66

Control group: routine preoperative diet (35 kcal/kg/day) and research group: 500 ml of EN suspension #

Data collected 48h within the first hospitalisation, the first day after NT and the first and 8th day after surgery

-

Higher BMI, serum PA, TP and ALB in trial group and a faster gastrointestinal recovery, shorter term use of drainage tubes, shorter hospital stay and less complications (p<0.05)

Preoperative EN and ALB were independent risk factors for PRNS. (P<0.05)

Lower NRS2002 and PGSGA in the trial group (p<0.05).

Claudino et al.

Brazil

2019

cohort

(retrospective)

Gastric cancer.

Subtotal or total gastrectomy.

Patients who did or did not undergo NAC

n=164

Two groups:

-immunonutrition¥ before surgery

-conventional

No significant difference in OS rates at 6-months, 1-year and 5-years (no significant difference in OS rates at 6-months (92.6% versus 85.0%; p=0.154) 1-year (87.0% versus 78.5%; p=0.153) and 5-years (69.6% versus 58.3%; p=0.137).

A trend for longer OS was found in immunonutrition group.

Immunonutrition group with less infectious complications (non-significant)

Immunonutrition group with less readmissions for surgical complications (non- significant) (41.1% vs 48.1%; p=0.413)

Zhao et al

China

2018

Randomized clinical trial

Locally advanced gastric cancer; NAC;

n=106

ERAS group or standard care group.

Sarcopenic patients had lower OS than non-sarcopenic patients (p<0.05).

No significant differences in OS for patients who experienced DLT.

BMI and BSA were lower in sarcopenic patients and associated with DLT.

-

Legend: NAC – neoadjuvant chemotherapy; LA – locally advanced,  GEJ – gastroeosophageal junction; DLT – dose limiting toxicity; NAT – neoadjuvant treatment; CT – chemotherapy; VAT - visceral adipose tissue; SAT - subcutaneous adipose tissue; DFS – disease free survival ; MN – malnutrition;  BMI – Body mass Index; PA - Prealbumin PRNS -Prognostic related nutritional score; mSIS - modified systemic inflammation score; CT Scan - Computed Tomography scan; RECIST - Response Evaluation Criteria in Solid Tumors; FLOT - fluorouracil plus leucovorin, oxaliplatin, and docetaxel; PGSGA - Patient Generated Subjected Global Assessment; SMI – skeletal muscle index; CONUT – Controlling Nutritional status

# Nutrison fiber- and oral nutritional supplementation (500 ml per bottle containing 500 kcal, 20 g protein, 19,45 g fat and 61,5 g CH); 7-days before surgery apart from routine preoperative diet (35 kcal/kg/day).

Both groups on Nutrison fiber within 48h after surgery

¥ Immune enteral diet enriched with arginine, omega-3 fatty acids and nucleotides.

$ ERAS group: sufficient preoperative patient education, normal diet until 6 hours before surgery, liquid intake until 2 hours before surgery, preoperative carbohydrate loading before surgery, analgesia with nonsteroidal anti-inflammatory drugs, minimization of opioid pain management, avoidance of perioperative fluid overload, no routine use of NGT, no abdominal drains, early removal of bladder catheters, liquid diet on recovery from anaesthesia, semi-liquid diet on return of bowel function, tolerated liquid diet and forced ambulation on the day of the surgery; NGT placed preoperatively and remained until flatus occurred, intra-abdominal drains placed during surgery until the day before discharge, not allowed oral intake until bowel flatus gastrointestinal movement occurred, usually remained in bed for approximately 2 days after surgery. Conventional group: gastrointestinal preparation before surgery, fasted from midnight.

3) The paper would benefit from a more thorough discussion of the limitations of the included studies. For example, the authors should discuss potential biases and confounding factors that may have influenced the results of the individual studies, as well as the overall conclusions of the review.

Thank you for your question. We agree with the concerns raised, and have, therefore, added in the manuscript information regarding the studies limitations.

Briefly, we have added in the manuscript: “Most of the studies found and included in this review had a retrospective design and recruited small sample size (single center). Many have also identified the following limitations: heterogeneous clinical data, inconsistencies in the prescribed treatment plan, time to follow-up and diverse cut-off values (Appendix 2 - Supplementary data).”: manus line 431.

Appendix 2 - Limitations of the included studies

Study

Limitation

Palmela et al

Portugal

2017

Retrospective design, single-centre recruitment, and small sample size.

Lack of staging laparoscopy. Some of the patients included in the study might have already had peritoneal disease, and this may explain the notable percentage of patients with disease progression during NAC-

The use of different regimens of CT drugs according to patient characteristics.

Yamaoka et al

Japan

2014

Retrospective study performed in a single institution. Relatively small size of the studied population warrants further studies.

Tan et al

UK

2015

Study with limited power paving the way for further evaluations in larger samples. BSA-based dosing for CT.

Zhang et al

China

2021

Single-centre retrospective study, with deficient sample size and follow-up time. Cut-off values are in line with current guidelines for sarcopenia diagnosing, but cut-off points for VFA and SFA have not been established.

Hand-grip strength and usual gait speed should be measured in addition to muscle mass in further research.

Zhou et al

China

2020

Due to the lack of an established cut-off point for low skeletal muscle and adipose mass, the first-quartile values for low skeletal muscle and adipose mass were used; usage of this cut-off point, to define low skeletal muscle mass, has been increasingly considered a good threshold indicator. Small retrospective study sample size warranting further investigation, to be further confirmed in prospective studies. Follow-up time was short requiring longer periods in future studies.

Zhang et al

China

2021

Retrospective single-centre study with a short follow-up time and small sample size. The number of patients with marked FAT loss was limited, which might have caused bias. The cut-off values for body composition changes during neoadjuvant treatment varied between the studies. Therefore, the optimal cut-off values for muscle mass loss, VFA loss, SFA loss and body weight loss were determined by X-tile plots. Some patients were not assessed for the pathologic response to neoadjuvant treatment; thus this factor was not included in the final model for survival.

Jiang et al

China

2021

Retrospective single-centre study with small sample size limits the ability to draw broader conclusions. The NAC is SOX, preventing further discussions with other CT option plans. Study included patients from one single academic medical centre, and therefore the outcomes may not be generalizable.

Rinninela et al

Italy

2021

Study with different cut-off to define muscle mass loss which thresholds depends on the characteristics of the studied population such as age, race and country. Secondly, most of the included studies were performed on Asian patients given the high prevalence of gastric cancer in Asia. Although we included data from the only available European studies, more research is needed to confirm our findings in non-Asian populations with gastric cancer.

Jin et al

China

2021

Retrospective single-centre study with small sample size limits the ability to draw broader conclusions. ROC curve for the pre-treatment CONUT score cut-off value was associated with a poor sensitivity.

Li et al

China

2020

Patients included in the analysis might have different plan treatments, such as in nutrition support, CT plans and CT cycles, which could affect immunity and nutritional status. Failing to incorporate this information might have biased results. Length of follow-up is relatively short and therefore long-term survival cannot be assessed. The nomogram developed has not been validated internally or externally, which would cast doubts on its generalizability.

Sun et al

China

2016

Retrospective single-centre study.

Zhao et al

China

2018

The effect of ERAS programs on patients who received NAC was not observed; differences between NAC patients and those without-NAC may be more significant than what was explored in this study. Single-centre clinical trial, and results from other centres are required. Long-term survival rate was not determined and, therefore, a follow-up to evaluate whether NAC in ERAS programs are benefit on the long‑term survival is necessary.

Claudino et al.

Brazil

2019

Lack of comprehensive data on dietary intake, given the nature of the retrospective study.

Legend: NAC – neoadjuvant chemotherapy; CT – chemotherapy; VFA- visceral fat adiposity; SFA - subcutaneous fat adiposity; FAT – fat adipose tissue; SOX  - S-1 and oxaliplatin; CONUT – Controlling Nutritional status  

4) It would be helpful to elaborate on the potential mechanisms underlying the associations between nutritional status, body composition, and clinical outcomes in gastric cancer patients. This would provide a more comprehensive understanding of the topic and guide future research in this area.

And

5) The authors should discuss the clinical implications of their findings, including any potential recommendations for the assessment and management of nutritional status in gastric cancer patients undergoing neoadjuvant treatment.

And

6) The conclusion section should be revised to provide a more focused summary of the main findings, as well as to address the implications for clinical practice and future research. 

Thank you ever so much for all your remarks because it gave us the opportunity to further improve our manuscript. Regarding your comment 4), 5) and 6) we have now proceeded to an in-depth reorganization of the manuscript and taken your comments to enrich our proposed discussion and conclusion. Please find all the alterations in the manuscript with track change. We hope to have been true to all the raised concerns.

Minor points:

  • The manuscript contains several grammatical and typographical errors, which should be corrected for better readability and clarity.

 Thank you for the comment. We agree and have changed in the manuscript accordingly.

  • Ensure consistency in the use of abbreviations throughout the manuscript, such as neoadjuvant chemotherapy (NAC) and overall survival (OS).

Thank you for the comments. We agree and have changed in text and tables accordingly.

3) In several instances, the paper uses an excessive amount of direct quotes from the reviewed studies. Consider paraphrasing these passages and summarizing the findings in the authors' own words to improve readability and reduce redundancy.

Thank you for the comment. We agree and hopefully with the addition of the new tables and the in-depth reorganization this is now acknowledged.

4) The formatting of the references should be checked to ensure they adhere to the journal's guidelines.

Thank you for the comment. We have checked and corrected accordingly.

Overall, the submitted paper provides valuable insights into the nutritional-related critical points in gastric cancer neoadjuvant management and their associations with clinical outcomes. Addressing the major and minor points outlined above will improve the quality, clarity, and impact of the manuscript.

Reviewer 2 Report

The article “Neoadjuvant gastric cancer treatment associated nutritional critical points for the optimization of care pathways: a systematic review”is well written review of the literature. Below I put some of my questions/remarks:

1.       Some abbreviations has been introduced in abstract without explanation e.g. NAC.

2.       Why search was restricted to the period after 2011? Could you justify in manuscript such choice?

3.       Only 151 articles were found after using the search strategy defined by Authors? Could you justify the search stratefy used? Don’t you think that the part of search strategy: (Nutritional status OR Nutritional intervention OR Nutritional support OR Dietary counselling OR Oral nutritional supplements)) is extremely restrictive and that the results may not include all articles adressing the problem?

4.       Don’t you think that the part “Tan et al. [17] showed a median OS for sarcopenic patients of 569 days (IQ range: 357-1230 days) and for patients who were not sarcopenic 1013 days (IQ range: 496-1318 days) (log-rank test, p=0.04). – Why this is put in this part not in sarcopenia?” should be placed in previous paragrapg: “3.1.2. Sarcopenia (baseline, pre-treatment)”?

5.       However, they found no significant difference in overall survival in patients who experienced DLT compared with those that did not (810 days [IQ range: 323-1417] vs. 859 days [IQ range: 445e1269]; p=0.665)”. I think some  typo is in  “IQ range: 445e1269”?

Author Response

  1. Some abbreviations have been introduced in abstract without explanation e.g. NAC.

Thank you for the comment. We agree and have changed accordingly.

  1. Why search was restricted to the period after 2011? Could you justify in manuscript such choice?

Thank you for your question. We believe the chosen ten-year time period is representative of the literature, as it was at that time, in 2011, and following on from the early MAGIC trial (2006), that the value of NAC (perioperative chemotherapy) compared with surgery alone was confirmed by the FNCLCC and the FFCD multicenter phase III trial, which then contributed to both an increased interest on this therapeutic approach and in the field’s literature.

Therefore, where it used to read this: “Following on from the MAGIC [9] and the FFCD/FNCLCC trials [10] the use of ECF (epirubicin, cisplatin and 5-fluorouracil) or CF (cisplatin and 5-FU) respectively, are common”, now reads “Following on from the MAGIC [9] and the confirmatory FFCD/FNCLCC trials [10] the use of ECF (epirubicin, cisplatin and 5-fluorouracil) or CF (cisplatin and 5-FU) respectively, became common”.

(9)      Cunningham, D.; Allum, W.; Stenning, S.; Thompson, J.; Van de Velde, C.; Nicolson, M.; Scarffe, H.; Lofts, F.; Falk, S.; Iveson, T.; Smith, D.; Langley, R.; Verma, M.; Weeden, S.; Chua, Y. Perioperative Chemotherapy versus Surgery Alone for Resectable Gastroesophageal Cancer. N. Engl. J. Med. 2006, 355 (1).

(10)    Ychou, M.; Boige, V.; Pignon, J. P.; Conroy, T.; Bouché, O.; Lebreton, G.; Ducourtieux, M.; Bedenne, L.; Fabre, J. M.; Saint-Aubert, B.; Genève, J.; Lasser, P.; Rougier, P. Perioperative Chemotherapy Compared with Surgery Alone for Resectable Gastroesophageal Adenocarcinoma: An FNCLCC and FFCD Multicenter Phase III Trial. J. Clin. Oncol. 2011, 29 (13), 1715–1721. https://doi.org/10.1200/JCO.2010.33.0597.

  1. Only 151 articles were found after using the search strategy defined by Authors? Could you justify the search stratefy used? Don’t you think that the part of search strategy: (Nutritional status OR Nutritional intervention OR Nutritional support OR Dietary counselling OR Oral nutritional supplements)) is extremely restrictive and that the results may not include all articles adressing the problem?

Thank you for your question. We intended to narrow as much as possible the scope of this systematic review so that the critical domains and the outcomes could be more clearly identified. Nonetheless, we looked into the literature, and, to the best of our knowledge, we do not think that any relevant article is missing from this review. Furthermore, we have now included an example of our search strategy (Appendix 1 – Supplementary data) and attempted to better clarify our inclusion and exclusion criteria (Table 2).

  1. Don’t you think that the part “Tan et al. [17] showed a median OS for sarcopenic patients of 569 days (IQ range: 357-1230 days) and for patients who were not sarcopenic 1013 days (IQ range: 496-1318 days) (log-rank test, p=0.04). – Why this is put in this part not in sarcopenia?” should be placed in previous paragraph: “3.1.2. Sarcopenia (baseline, pre-treatment)”?

Thank you for your comment. We have changed accordingly.

  1. However, they found no significant difference in overall survival in patients who experienced DLT compared with those that did not (810 days [IQ range: 323-1417] vs. 859 days [IQ range: 445e1269]; p=0.665)”. I think some typo is in “IQ range: 445e1269”?

Thank you for your comment. We have changed accordingly.

Round 2

Reviewer 1 Report

The authors improved the text following my suggestions